DOI: 10.1038/ncomms15421　　OPEN

# Organic narrowband near-infrared photodetectors based on intermolecular charge-transfer absorption

Bernhard Siegmund[1,2], Andreas Mischok[1,2], Johannes Benduhn[1,2], Olaf Zeika[1,2], Sascha Ullbrich[1,2], Frederik Nehm[1,2], Matthias Böhm[2], Donato Spoltore[1,2], Hartmut Fröb[1,2], Christian Körner[1,2], Karl Leo[1,2] & Koen Vandewal[1,2]

Blending organic electron donors and acceptors yields intermolecular charge-transfer states with additional optical transitions below their optical gaps. In organic photovoltaic devices, such states play a crucial role and limit the operating voltage. Due to its extremely weak nature, direct intermolecular charge-transfer absorption often remains undetected and unused for photocurrent generation. Here, we use an optical microcavity to increase the typically negligible external quantum efficiency in the spectral region of charge-transfer absorption by more than 40 times, yielding values over 20%. We demonstrate narrowband detection with spectral widths down to 36 nm and resonance wavelengths between 810 and 1,550 nm, far below the optical gap of both donor and acceptor. The broad spectral tunability via a simple variation of the cavity thickness makes this innovative, flexible and potentially visibly transparent device principle highly suitable for integrated low-cost spectroscopic near-infrared photodetection.

[1] Dresden Integrated Center for Applied Physics and Photonic Materials (IAPP), Technische Universität Dresden, George-Bähr-Straße 1, Dresden 01062, Germany. [2] Institute for Applied Physics, Technische Universität Dresden, George-Bähr-Straße 1, Dresden 01062, Germany. Correspondence and requests for materials should be addressed to B.S. (email: bernhard.siegmund@iapp.de) or to K.V. (email: koen.vandewal@iapp.de).

Mixtures of organic semiconductors with displaced frontier orbital energies are applied in organic light-emitting diodes (OLEDs), solar cells and photodetecting devices. In molecular blends that allow for charge transfer, it is energetically favourable for an excited electron to reside on the acceptor molecule (A) and a hole to reside on the electron donating molecule (D)[1]. An encounter of both charges at a donor–acceptor interface results in the formation of an intermolecular charge-transfer (CT) state which can redissociate into free carriers or recombine to the ground state[2]. Their decay can be optimized for efficient emission as recently demonstrated for 'exciplex'-type OLEDs with a respectable external electroluminescence quantum yield of 15% (ref. 3).

Furthermore, CT states play an important role in organic photovoltaic and photodetecting devices, as they mediate between photo-generated excitons in neat absorbers and free charges[4]. As low-energy recombination centres, they limit the open-circuit voltage of organic solar cells[5,6] as well as the minimum achievable dark current in photodetectors[7]. Current research efforts in both device classes aim to synthesize absorber materials with sensitivity in the near-infrared (NIR) by lowering their optical gap[8]. However, despite several strategies[9–13], new organic materials rarely result in devices with reasonable responsivity above 1 μm (refs 14–17). Moreover, most organic donor–acceptor (D:A) blends offer a rather broadband absorption, while a tunable narrowband absorption would be of particular interest, for example, for spectroscopic applications[18,19].

Here, the use of direct intermolecular CT absorption could provide an elegant solution: as its absorption onset is determined by the energetic difference between the highest occupied molecular orbital (HOMO) of the donor and the lowest unoccupied molecular orbital (LUMO) of the acceptor, it can be substantially redshifted as compared to the absorption of neat materials[20]. However, due to its intermolecular nature, CT absorption is typically two orders of magnitude weaker than singlet absorption of the neat absorbers, leaving it often undetected[20]. To increase the CT absorption, ultrathick D:A blends of about 10 μm thickness have been suggested[21]. Due to a high series resistance, such a device can only be read out when applying extraction voltages in the order of 100 V. As a result of low on/off ratios, and presumably low speed, this approach to exploit CT absorption for photodetection is apparently abandoned. Therefore, in contrast to CT emission in 'exciplex' OLEDs, CT absorption is practically unused in organic optoelectronic devices up to now.

In this work, we introduce a resonant optical microcavity device which exploits the weak, but spectrally broadband CT absorption to achieve narrowband photodetection and tunable resonance wavelengths. We demonstrate a 41-fold enhancement of the external quantum efficiency (EQE) at the cavity resonance wavelength, with full-widths at half-maximum (FWHMs) down to 36 nm. At short circuit, we achieve EQEs exceeding 20% with photocurrent generation predominantly due to direct CT state excitation. A variation of the cavity thickness allows to tune the resonance wavelength in the CT absorption band over a vast range, from 810 to 1,550 nm, using a single D:A blend. We believe the original device concept introduced in this paper to open doors towards organic, flexible, cheap and ultracompact NIR spectrometers.

## Results

The resonant cavity device architecture and a simplified energy level diagram are shown in Fig. 1a,b. We utilize microcavities formed by a fully reflecting and a partially transmissive silver (Ag) mirror[22–24]. The simultaneous use as electrodes allows a

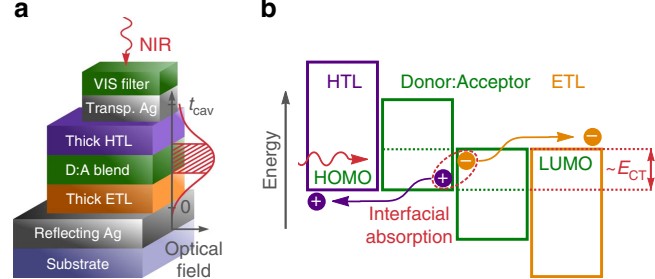

**Figure 1 | Working principle of the original cavity enhanced organic photodetector.** (**a**) Simplified scheme of the device architecture with a sketch of the optical field distribution for the resonance wavelength in the NIR. The thickness of both transport layers is chosen to achieve constructive interference in the CT absorption band and to situate the D:A blend layer in the maximum of the optical field. Layer thicknesses are not drawn to scale. (**b**) Simplified energy diagram at open circuit. A photon with less energy than the optical gaps and at least the CT state energy $E_{CT}$ is absorbed at the interface between an electron donating semiconductor and $C_{60}$ as acceptor. Thereby, an electron in an occupied state on the donor is promoted into an unoccupied state on the acceptor. The resulting intermolecular CT state dissociates into a free electron and hole which are extracted via the electron (ETL) or hole transport layer (HTL) at the respective Ag electrode.

compact design incorporating both optical and electrical elements. The photo-active D:A blend is sandwiched between transparent transport layers which allow selective charge extraction towards the outer electrodes, analogous to state-of-the-art organic photovoltaic diodes[25–28]. Within the Fabry–Pérot interferometer formed by the two Ag mirrors, electromagnetic waves with a wavelength $\lambda_{res}$ experience constructive interference when $\lambda_{res} = 2\hat{n}t_{cav}/j$, with $j$ being a natural number. The effective optical cavity thickness $\hat{n}t_{cav}$ is determined by the average index of refraction $\hat{n}$ and the effective geometrical cavity thickness $t_{cav}$, that is, the distance between both Ag mirrors extended by the field penetration into both metal layers[29,30].

To demonstrate the concept, we use a photo-active blend comprising buckminsterfullerenes ($C_{60}$) as electron acceptor and zinc phthalocyanine (ZnPc) as electron donor whose chemical structures are depicted in Fig. 2b. Their energy levels are displaced as illustrated in Fig. 1b. The ZnPc:$C_{60}$ blend is known to display a substantially redshifted CT state as compared to the neat absorbers[31]. A volume ratio of 1:1 provides a maximum contact interface between ZnPc and $C_{60}$. A sensitive EQE spectrum of a reference photovoltaic device with minimized cavity effects is depicted in Fig. 2a as crossed green line. The measurement confirms the presence of CT absorption in the NIR above 850 nm, that is, below the optical gap of neat $C_{60}$ (700 nm) and ZnPc (815 nm) whose absorption spectra are shown as grey lines marked as I and II, respectively.

**Cavity-enhanced CT absorption.** To make use of the broadband weak CT absorption, we greatly enhance the optical field in the NIR utilizing a resonant microcavity. The thicknesses of the transport and Ag layers are optimized via optical transfer-matrix simulations[32] for an absorber blend of 50 nm, as shown in Supplementary Fig. 1. To reach narrowband cavity resonances in the spectral region of CT absorption, we choose transport layer thicknesses of above 60 nm which exceeds those typically used in organic solar cells[25–27,33]. The EQE spectrum of a device fabricated with optimum transparent electrode (18 nm) and transport layer thicknesses (both about 80 nm, more details in Supplementary Table 1) is depicted in Fig. 2a on linear and

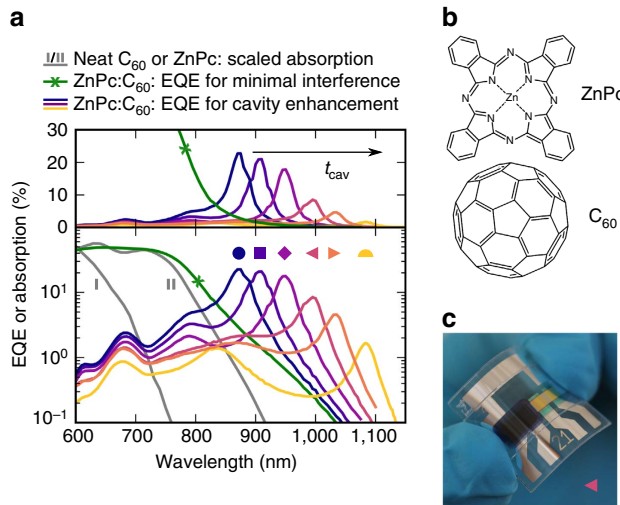

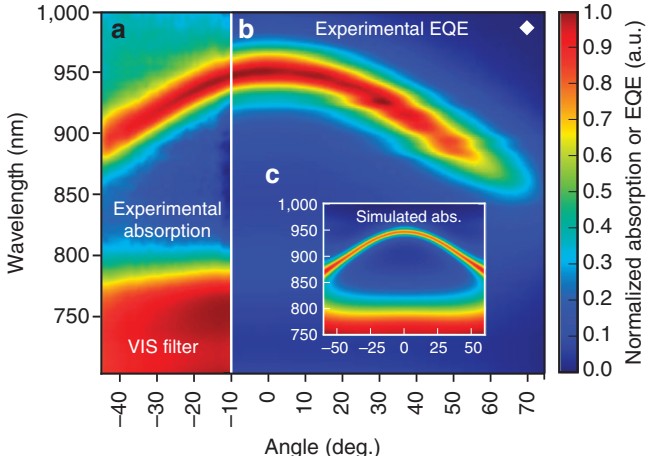

**Figure 2 | NIR detectors based on ZnPc:C$_{60}$ CT absorption. (a)** Spectrally resolved EQE or absorption on linear (top) or logarithmic (bottom) scale. The green, crossed line indicates the EQE of a ZnPc:C$_{60}$ solar cell with minimal optical cavity effect. The grey lines are scaled to the previous curve and represent the absorption of neat C$_{60}$ (marked as I) or ZnPc (marked as II). The remaining curves show the EQE of several cavity-enhanced ZnPc:C$_{60}$ detectors, measured at short circuit. Tuning the cavity thickness $t_{cav}$ via a simultaneous thickness variation of both transport layers allows shifting the resonance wavelength. Both samples marked as triangles are mechanically flexible, while the remaining are encapsulated with glass. **(b)** Chemical structures of the blend constituents. **(c)** Photograph of a flexible device, marked as ◄ in **a**. The left half of the photo-active area features a sequence of neat absorbers to suppress the photo-response from visible light.

**Figure 3 | Cavity-induced dependence of the resonance wavelength on the angle of light incidence.** Angularly and spectrally resolved behaviour of a ZnPc:C$_{60}$ sample resonating at 950 nm when the angle of incidence is 0° (device marked as ♦ in Fig. 2a). **(a)** Experimental device absorption. **(b)** Experimental EQE. **(c)** Simulated device absorption. All graphs are normalized. For wavelengths below 800 nm, the neat absorber sequence extinguishes light outside the microcavity (see **a,c**) and, therefore, efficiently reduces the photo-response in the visible spectrum (compare **b**).

logarithmic scale as purple line marked as ♦. The EQE reaches 18% at the resonance wavelength $\lambda_{res}$ of 950 nm with an FWHM of 36 nm. As compared to the reference device without cavity enhancement, we found a 27-fold increase in EQE at 950 nm. Moreover, a neat material filter sequence outside the cavity

quenches the photo-response stemming from strong absorption above the optical gap of ZnPc, as shown in Fig. 1a and Supplementary Fig. 2.

**Resonance wavelength tuning**. To demonstrate the tunability of the detection wavelength $\lambda_{res}$, we vary the thickness of both transport layers between 60 and 105 nm, resulting in a spectral shift of the cavity resonance. As shown by the EQE spectra in Fig. 2a, we obtain resonances distributed over more than 200 nm within the CT band of ZnPc:C$_{60}$. A maximum EQE of 23% is achieved for a resonance wavelength of 875 nm, as depicted by the dark blue line marked as ●. On increasing cavity thickness, the resonance wavelength shifts from 875 to 1,085 nm with increasing EQE amplification up to 41 times (for more details see Supplementary Fig. 3). While the EQE decreases for longer wavelengths, the large cavity enhancement still guarantees values above 1%, even 380 meV below the optical gap of neat ZnPc. Furthermore, to demonstrate an important asset of organic devices, two samples, with a detection wavelength of 995 nm (◄) and 1,035 nm (►), are deposited onto a flexible substrate. Exemplary, one fully flexible encapsulated device is depicted in Fig. 2c. To allow for further device integration, we demonstrate photodetectors with visible transparency by shaping both electrodes partially translucent, as shown in Supplementary Fig. 8.

**Angular dependence of resonance**. Since the device class introduced here is based on optical interference, the cavity enhancement depends on the angle of light incidence. Figure 3 shows the absorption (3a) and EQE spectra (3b) of a device resonating at 950 nm, marked as ♦ in Fig. 2a, as a function of the angle of incidence with respect to the substrate normal. The resonance wavelength decreases for non-normal excitation, with the peak position following a parabolic dispersion, leading to a reduction in resonance wavelength of more than 50 nm for angles above 45°. As illustrated in Fig. 3c, this behaviour is reproduced by transfer-matrix simulations. Close to normal incidence, there is however a rather broad angular range, spanning from −20° to +20°, where the resonance shift does not exceed 10 nm which is substantially smaller than the FWHM of 36 nm.

**Identification of parasitic absorption**. In the reported devices, the peak EQE is currently limited by parasitic absorption, which decreases the number of photon transits through the D:A blend. This observation is evidenced in Fig. 4a depicting the experimentally measured device absorption (filled area) and the corresponding EQE (hatched area) of three previous ZnPc:C$_{60}$ devices whose EQEs peak at 910, 950 and 995 nm, respectively. While the peak EQE significantly drops (−59%) with increasing $\lambda_{res}$, the device absorption remains almost unchanged (−3%). Therefore, parasitic absorption greatly exceeds CT absorption. More insight into the origin of the parasitic absorption at the cavity resonance is obtained from optical simulations, depicted in Fig. 4b. We identify the transmissive top electrode, where almost every second photon absorption occurs, as main responsible. Moreover, the transparent and reflective electrode together yield the highest parasitic contribution with between 60 and 80% of the resonance absorption, followed by the doped N4,N4′-bis(9,9-dimethyl-9H-fluoren-2-yl)-N4,N4′-diphenylbiphenyl-4,4′-dia-mine (BF-DPB) hole transport layer with about 15%.

**Longer detection wavelengths**. While providing the proof of principle, ZnPc:C$_{60}$ has a CT absorption band limited to wavelengths not exceeding 1,100 nm. To detect photons with lower energies, we exchange ZnPc by 2,2′,6,6′-tetraphenyl-4,4′-bipyr-anylidene (TPDP), chemical structure in Fig. 5b) as donor with an

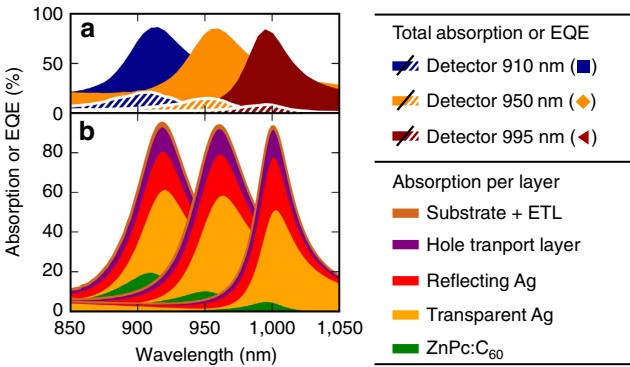

**Figure 4 | Identification of parasitic absorption at the resonance wavelength.** Three ZnPc:$C_{60}$ devices from Fig. 2a with resonances at 910 nm (■), 950 nm (◆), and 995 nm (◀) are analysed. (**a**) The experimental device absorption is shown as filled area and the corresponding EQE as hatched area. (**b**) The simulated absorption is estimated for each layer. The legend entries are given in the order of appearance.

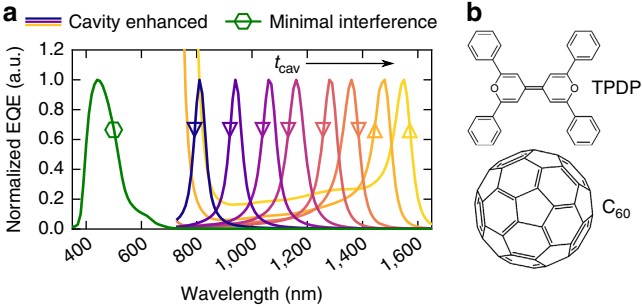

**Figure 5 | NIR detectors based on TPDP:$C_{60}$ CT absorption.** (**a**) Normalized EQE spectra of several tetraphenyl dipyranylidene:fullerene (TPDP:$C_{60}$) detectors at short circuit. The reference sample (○) with minimal interference effects has negligible absorption for wavelengths above 700 nm. For the cavity-enhanced devices, the resonance wavelength is varied from 810 to 1,550 nm by a thickness variation of both transport layers (▽) — and for selected detectors also by a thickness variation of the TPDP:$C_{60}$ blend (△). (**b**) Chemical structures of the blend constituents.

elevated HOMO level[34] which reduces the onset energy of CT absorption. A sensitive EQE measurement of a TPDP:$C_{60}$ solar cell with minimal interference effects reveals a remarkably broad and featureless CT band from 725 nm down to 1,600 nm, as shown in Supplementary Fig. 4. Utilizing the approach outlined above, we tune the cavity by thickness adjustments of both transport layers (marked as [triangle down]) — and for selected devices also by thickness variations of the D:A blend (△). For more details on the layer thicknesses used, we refer the reader to Supplementary Table 2. Following this strategy, we achieve a remarkably broad resonance tunability from 810 to 1,550 nm as shown in Fig. 5a. We like to emphasize that the latter detection wavelength, addressing CT states 1 eV below the optical gap of both neat $C_{60}$ and TPDP, settles among the highest wavelengths achieved with organic photodetectors so far[14–16]. Up to now, we observe rather low internal quantum efficiencies for the material blend TPDP:$C_{60}$, decreasing with blend thickness from 20 to 3%. However, we expect future alternative D:A combinations to achieve higher internal yields for CT excitation in this wavelength range.

## Discussion

The device concept outlined in this work has great potential as an original class of organic, narrowband NIR detectors, with an easily tunable detection wavelength. It is worth emphasizing that the wavelength selectivity, provided by these devices, is explicitly due to the weakly absorbing nature of CT states: in contrast, more strongly absorbing neat material transitions do not allow such a strong cavity enhancement and narrowband resonances as observed here, except within their narrow absorption tail region (for more details see the Supplementary Discussion)[22,35]. Moreover, the outlined detection principle undergoes a paradigm change: while conventionally excitons diffuse[36] from either the absorbing donor or acceptor to a joint D:A interface to decay into CT states, those steps are skipped here in favour of a direct absorption of the latter. Demonstrating detectors with EQEs around 20%, this work underlines experimentally that CT states can be rather efficiently converted into photocurrent—which is contrary to their previous perception as trap states[37]. Given the achieved dark currents, which are still limited by extrinsic device shunts, we estimate an upper limit for the specific detectivity at the resonance wavelength of the ZnPc:$C_{60}$-based detectors of $10^{11}$ Jones at short circuit (details are provided in the Supplementary Methods and Supplementary Fig. 5). We expect a further reduction in noise current and enhancement in detectivity by optimizing the absorber layer thickness or introducing undoped blocking layers.

On a variation of the excitation intensity, we observe no deviation from a linear photo-response over more than 5 orders of magnitude, as shown in Supplementary Fig. 6. We measure rise and fall times (10 to 90% or reverse) of 3 and 151 ns, respectively (see Supplementary Fig. 7 and Supplementary Table 1). The latter is partly delayed by discharge dynamics[38] of a resistor–capacitor circuit of a 0.25 mm$^2$ large device which might be accelerated by reducing the device area. However, this response time is already sufficiently short for numerous applications related to NIR photodetection such as contact-free movement detection, non-invasive subsurface vision or night vision[9].

A further increase in EQE and specific detectivity is expected when improving the ratio between CT absorption and the competing parasitic absorption. A first group of approaches aims to enhance the interfacial absorption, for example, by increasing the D:A blend thickness or by exploiting intercalating D:A blends with rather high CT absorption coefficients[39,40]. A second strategy consists of reducing the competing amount of parasitic absorption. As discussed in Fig. 4, the EQE of the presented ZnPc:$C_{60}$ devices can be improved via transport layers with suppressed NIR absorption. A much more drastic enhancement in EQE height and FWHM is expected when replacing the conducting Ag mirrors, being the dominant source for parasitic absorption: for this purpose, low loss mirrors such as distributed Bragg reflectors[41] or other high-quality resonators[42,43], paired with NIR transparent electrodes with a high in-plane conductivity would offer a promising perspective.

A multitude of NIR applications in biomedicine, pharmacy and agriculture relies on spectroscopic analysis—such as disease detection[44], determining blood concentrations of glucose, oxyhemoglobin and water[9,45], analysing and interacting with brain functions[46,47]; raw material and on-line quality monitoring[48]; or determining nutrient compositions and optimal harvest dates[49,50]. Following previously reported approaches, the realization of an organic NIR spectrometer would require several different donor or acceptor materials with varying optical gaps[18,51]. The replacement of the neat material extinction by the interfacial CT absorption which extends over several hundreds of nanometers provides an elegant, robust and cheap alternative: here, all detection wavelengths within the spectrometer range can be addressed solely by a thickness variation for a single D:A blend, as outlined in Fig. 5a. Especially for the analysis of chemical compositions as common task for

NIR spectroscopy, an even further extension of the detection wavelengths into the infrared would be desirable. Hereby, the outlined strategy will provide photosensitivity also beyond 1,550 nm on a proceeding reduction of the CT absorption onset, by an appropriate choice of the frontier energy levels of both D and A. However, using organics, we expect a detection limit at about 2 μm. Here, fundamental interatomic vibration modes will cause strong parasitic absorption[9] and, consequently, reduce the number of photon transits in the resonator.

In summary, we introduce an innovative class of organic narrowband NIR photodetectors based on mixtures of $C_{60}$ and donor materials with a high HOMO level. An optical cavity device architecture enhances the photocurrent for wavelengths within the intermolecular CT absorption band. Using mixtures of $ZnPc:C_{60}$, we obtain narrowband photodetection at wavelengths below the optical gap of ZnPc and $C_{60}$ with EQEs of above 20% and spectral widths down to 36 nm. For photodetectors based on $TPDP:C_{60}$ blends with a lower CT absorption onset, we demonstrate a tunability of the resonance wavelength over a strikingly broad range from 810 to 1,550 nm by simple variations of the cavity thickness. We believe that, due to its mechanical flexibility, light weight, scalability, low fabrication cost and potential transparency at visible wavelengths, the introduced device class will become a valuable candidate for integrated spectroscopic NIR photodetection.

## Methods

**General fabrication procedure.** Precleaned glass is used either as a neat rigid substrate or with a prestructured layer of 90 nm indium tin oxide (ITO; Thin Film Devices, USA) deposited on top. Flexible devices are processed onto 125-μm-thick films of planarized polyethylene naphthalate (pPEN; Teonex (R) PQA1M, DuPont Teijin Films, UK). Before device deposition, the flexible pPEN substrates are covered with 20 nm of aluminium oxide ($AlO_x$) as gas barrier by means of plasma-enhanced atomic layer deposition (Sentech SI ALD LL, Sentech Instruments, Germany), as earlier reported in ref. 52.

The subsequent layers composed of organics, fullerene, oxides and/or metals are deposited via thermal evaporation under controlled vacuum with a base pressure of $10^{-8}$ mbar (K.J. Lesker, UK). Evaporation rates, layer thicknesses and, where applicable, mixing ratios are controlled via quartz crystal microbalances, with rates not exceeding 1 Å s$^{-1}$. The geometrical intersection of the bottom Ag or ITO electrode and the top Ag or aluminium electrode defines a photo-active area of 6.4 or 0.25 mm$^2$.

After evaporation, all glass samples are covered with special encapsulation glasses which leave a sealed hollow volume filled with nitrogen above the device. A ultraviolet-cured epoxy glue (XNR 5592; Nagase ChemteX, Japan) is used to seal the sample at the rim of the encapsulation glass. All samples on flexible substrates are sealed with another flexible barrier film against oxygen and moisture. For this, a pPEN substrate with predeposited 20 nm atomic layer deposition $AlO_x$ is laminated (full area) onto the device. This is done using a 25-μm-thick, ultraviolet-cured, proprietary barrier glue (Tesa SE, Germany), containing a latent getter. The lamination is carried out at room temperature in inert atmosphere. $AlO_x$ films of both flexible barriers are placed directly adjacent to the device to minimize edge diffusion.

**ZnPc:$C_{60}$ series.** The layer sequences of all $ZnPc:C_{60}$ samples are documented in Supplementary Table 1. For further details on the device structure, we refer to Supplementary Note 1. Mass-related mixing ratios are given in weight per cent (wt%), whereas unitless ratios are volume specific. The utilized materials in order of appearance are: molybdenum trioxide ($MoO_3$; Sigma-Aldrich, USA); gold (Au; Allgemeine Gold und Silberscheidanstalt, Germany); Ag (K.J. Lesker); N,N-Bis(-fluoren-2-yl)-naphthalenetetracarboxylic diimide (IAPP, Germany; ref. 25); tetrakis (1,3,4,6,7,8-hexahydro-2H-pyrimido[1,2-a]pyrimidinato)ditungsten(II) (Novaled, Germany); 4,7-diphenyl-1,10-phenanthroline (BPhen; Lumtec, Taiwan); cesium (Cs; SAES Getters, Italy); buckminsterfullerenes ($C_{60}$, LUMO at 4.0 e.V; ref. 53; CreaPhys, Germany); ZnPc (HOMO at 5.1 eV; ref. 54; TCI Europe N.V., Belgium); BF-DPB (Synthon, Germany); NDP9 (proprietary p-dopant; Novaled); 2,2′-(perfluoronaphthalene-2,6-diylidene)dimalononitrile ($F_6$-TCNNQ; Novaled); tris-(8-hydroxy-quinolinato)-aluminium (Sigma-Aldrich, USA); and 2,3,10,11-tetrapropyl-1,4,9,12-tetraphenyl-diindeno[1,2,3-cd:1′,2′,3′-lm]perylene (IAPP). All organic materials expect the n-dopants are purified at least once by vacuum gradient sublimation.

As shown in Supplementary Table 1, a $ZnPc:C_{60}$ reference device with minimal optical interference uses an ultra-thin, highly transparent Ag electrode and transport layers thinner than 40 nm to resonate at the peak extinction of ZnPc

above its optical gap. Further cavity-enhanced $ZnPc:C_{60}$ photodetectors are built utilizing an alternative hole transport layer with $F_6$-TCNNQ as disclosed p-dopant. As shown in Supplementary Fig. 8, the formation and spectral shift of resonance peaks can be reproduced. Moreover, a reference device with neat $C_{60}$ as photo-active layer, marked as I, follows the layer sequence Glass|ITO|$MoO_3$|$C_{60}$|BPhen|Ag. A further neat ZnPc reference device, marked as II, follows the layer sequence Glass|ITO|$C_{60}F_{36}$|BF-DPB:$C_{60}F_{36}$|ZnPc|BPhen|Al.

**TPDP:$C_{60}$ series.** TPDP is synthesized from dimerizing pyrylium salts via a Wittig reaction as described in ref. 55. The layer sequence of all $TPDP:C_{60}$ devices is documented in Supplementary Table 2.

**Measurement techniques.** For details on the measurements techniques, we refer the reader to the Supplementary Methods.

**Data availability.** The data that support the findings of this study are available from the corresponding author on reasonable request.

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

## Acknowledgements

We thank the 'Bundesministerium für Bildung und Forschung' for funding within the scope of the projects InnoProfile 2.1 (03IPT602A) and InnoProfile 2.2 (03IPT602X), as well as the 'Deutsche Forschungsgemeinschaft' within the scope of SPP1839 (LE747/53-1). Moreover, we thank Annette Petrich for purification of the organic materials as well as Daniel Schütze, Andreas Wendel, Tobias Günther and Caroline Walde for sample preparation. We also thank Dr Mauro Furno (Novaled GmbH) and Sim4tec for providing the software for optical transfer matrix calculations. K.L. acknowledges his support as fellow of the Canadian Institute for Advanced Research (CIFAR).

## Author contributions

B.S. and K.V. developed the concept and conceived the experiments with A.M., J.B., D.S. and C.K. Moreover, A.M. designed the mirrors and optically optimized the microcavities including simulations of suitable layer thicknesses. O.Z. synthesized the donor material TPDP. F.N. provided flexible substrates and encapsulation. B.S. and J.B. measured and optimized the device EQE. B.S., A.M. and M.B. measured or simulated the angle-dependent device characteristics. B.S. determined the experimental and simulated absorption under normal incidence. S.U. and D.S. measured the transient photocurrent. B.S. carried out measurements of the dynamic range, current–voltage characteristics, noise and specific detectivity. B.S. wrote the manuscript with feedback from K.V. Beyond, H.F., K.L. and K.V. supervised parts of the research. K.V. was the project leader. All authors contributed to discussing the results and finalizing the paper.

## Additional information

**Competing interests:** The authors declare no competing financial interests.

**Publisher's note**: 

