## [Peer Review File · Nature Communications]

Reviewers' comments:

Reviewer #1 (Remarks to the Author):

The authors have substantially revised the manuscript. The results are exciting and important for the development of near IR photo-detectors. I believe that, the revised manuscript is within the scope of this journal and should be published without delay.

Reviewer #2 (Remarks to the Author):

The authors addressed in detail all my concerns and questions. I do appreciate the care in each answer.

Upon revision, I cannot but confirm the relevance of this work in successfully proposing an alternative strategy for NIR detection with organics, which may foster further research indeed. With the extended discussions and clarifications added in the work, it will be easier for the community to realize the unique features characterizing amplified intermolecular CT absorption. The path for improvement of efficiencies at long wavelengths and to achieve visible blindness is better outlined and plausible.

I do recommend the publication of this work in Nature Communications.

The authors may consider the following text amendments:

- After a semi-column, a lower case letter should be used, not a capital letter (lines 34, 187, 190, 216, 228)
- Line 42: replace "100 Volts" with "100 V"
- Line 43: "CT absorption for photodetection was abandoned". Since you never know, I'd like to suggest to write "CT absorption for photodetection is apparently abandoned".

1st Review

Quotation 1: The authors have substantially revised the manuscript. The results are exciting and important for the development of near IR photo-detectors. I believe that, the revised manuscript is within the scope of this journal and should be published without delay.

Reply 1: We are pleased about the very positive feedback from the referee confirming the relevance and novelty of this work. Moreover, we thank him/her once more for the constructive comments from the previous iteration which helped us to strengthen this work.

2nd Review

Quotation 1: The authors addressed in detail all my concerns and questions. I do appreciate the care in each answer. Upon revision, I cannot but confirm the relevance of this work in successfully proposing an alternative strategy for NIR detection with organics, which may foster further research indeed. With the extended discussions and clarifications added in the work, it will be easier for the community to realize the unique features characterizing amplified intermolecular CT absorption. The path for improvement of efficiencies at long wavelengths and to achieve visible blindness is better outlined and plausible. I do recommend the publication of this work in Nature Communications.

The authors may consider the following text amendments:

- After a semi-column, a lower case letter should be used, not a capital letter (lines 34, 187, 190, 216, 228)
- Line 42: replace “100 Volts” with “100 V”
- Line 43: “CT absorption for photodetection was abandoned”. Since you never know, I’d like to suggest to write “CT absorption for photodetection is apparently abandoned”.

Reply 1: We are pleased about the very positive feedback from the referee confirming the relevance and novelty of this work. Moreover, we thank him/her once more for the constructive comments from the previous iteration which helped us to strengthen this work. Moreover, we followed the referees suggestion raised in the present correspondence leading to a couple of minor changes in the main text as displayed below:

- "the use of direct intermolecular CT absorption could provide an elegant, novel solution: as its absorption onset is determined by the energetic difference [...]" (Line 34)
 - "the wavelength selectivity [...] is explicitly due to the weakly absorbing nature of CT states: in contrast, more strongly absorbing neat material transitions [...]" (Line 187)
 - "the outline detection principle undergoes a paradigm change: while conventionally, excitons diffuse from either the absorbing donor or acceptor [...]" (Line 190)
 - "A much more drastic enhancement in EQE height and FWHM is expected when replacing [...]: for this purpose, low loss mirrors [...]" (Line 216)
 - "The replacement of the neat material extinction [...] provides an elegant, robust and cheap alternative: here, all detection wavelengths within [...]" (Line 228)
 - "Due to a high series resistance, such a device can only be read out when applying extraction voltages in the order of 100 V. As a result of low on/off ratios, and presumably low speed, this approach to exploit CT absorption for photodetection is apparently abandoned." (Line 42-43)
-